# Severe COVID-19 with persistent respiratory failure—A retrospective cohort study in a tertiary centre in Malaysia

**Chee Kuan Wong**[1⊙], **Leng Cheng Sia**[1⊙]*, **Noreen Zhi Min Ooi**[2], **Wai Yee Chan**[3], **Yong-kek Pang**[1]

**1** Department of Medicine, University Malaya Medical Centre, Kuala Lumpur, Malaysia, **2** Ministry of Health, Kuala Lumpur, Malaysia, **3** Imaging Department—Gleneagles Hospital Kuala Lumpur, Kuala Lumpur, Malaysia

⊙ These authors contributed equally to this work.
* lcsia@ummc.edu.my

## Abstract

### Introduction

Management of severe COVID-19 patients with persistent respiratory failure after acute phase treatment is not only challenging, but evidence for treatment is scarce, despite some authors reporting favourable clinical responses to corticosteroid therapy in histologically proven secondary organising pneumonia (OP). This study aimed to report the course of the disease, radiological pattern and clinical outcomes of severe COVID-19 patients with persistent respiratory failure.

### Methods

This was a retrospective cohort study of severe COVID-19 patients who were admitted to a single tertiary centre from 1 January 2021 to 30 June 2021. The clinical data of the patients during admission and clinic follow-up, including radiological images, were traced using electronic medical records.

### Results

In our cohort, the mortality rate for those with severe COVID-19 was 23.1% (173/749). Among the survivors, 46.2% (266/576) had persistent respiratory failure (PRF) after 14 days of illness. Of them, 70.3% (187/266) were followed up, and 68% (128/187) received oral corticosteroid (prednisolone) maintenance treatment. OP pattern made up the majority (81%) of the radiological pattern with a mean severity CT score of 10 (SD±3). The mean prednisolone dose was 0.68mg/kg/day with a mean treatment duration of 47 days (SD±18). About one-third of patients (67/187) had respiratory symptoms at 4 weeks (SD±3). Among 78.1% (146/187) who had a repeated CXR during follow-up, only 12 patients (8.2%, SD±3) had radiological improvement of less than 50% at 6 weeks (SD±3), with 2 of them later diagnosed as pulmonary tuberculosis. Functional assessments, such as the 6-minute walk test and the spirometry, were only performed in 52.4% and 15.5% of the patients, respectively.

**Data Availability Statement:** All relevant data are within the article and its Supporting Information files.

**Funding:** The author(s) received no specific funding for this work.

## Conclusion

Almost half of the patients with severe COVID-19 had PRF, with a predominant radiological OP pattern. More than two-thirds of the PRF patients required prolonged oral corticosteroid treatment. Familiarising clinicians with the disease course, radiological patterns, and potential outcomes of this group of patients may better equip them to manage their patients.

## Introduction

The pandemic caused by severe acute respiratory syndrome-coronavirus 2 (SARS-CoV-2) or coronavirus disease 19 (COVID-19) has infected more than 460 million of the world population with at least 6 million deaths as of 6 May 2022 [1]. The worldwide daily report remained high after two years since it was first reported as an outbreak in Wuhan, China, in December 2019 [2]. COVID-19 manifestation can range from mild to severe illness. The mortality rate of severe COVID-19 admitted to intensive care units has been reported as high as 35.5% in a meta-analysis [3].

Increasingly, reported cases of patients surviving severe COVID-19 pneumonia with persistent respiratory failure were attributed to secondary organising pneumonia. Short-course systemic corticosteroids were given to this group of patients with good outcomes [4–6]. A study by Myall et al. indicated that there might be a role of corticosteroids in post-discharge patients with COVID-related interstitial lung disease (ILD) [7]. However, this cohort was not highlighted in most studies despite the observed trend worldwide. The British Thoracic Society recommends a clinical consultation at 4 weeks followed by a repeat chest x-ray (CXR) at 12 weeks in follow-up for severe COVID-19 patients [8]. Malaysia adopts a similar follow-up protocol to detect potential complications such as pulmonary embolism, superadded infection, interstitial lung disease, pneumothorax, cardiac events etc [9].

Hence, this study aimed to observe the disease course, radiological pattern and clinical outcome of severe COVID-19 patients with PRF.

## Methods

This was a retrospective cohort study of adults with severe COVID-19 and persistent respiratory failure who tested positive for SARS-CoV-2 by a reverse transcriptase polymerase chain reaction assay in their nasopharyngeal and/or oropharyngeal swabs. All severe COVID-19 patients admitted to a single tertiary centre in Kuala Lumpur, Malaysia, from 1 January to 30 June 2021, were included in the study. The clinical data of the patients during admission and clinic follow-up, including radiological images, were traced using the hospital's electronic medical records. University Malaya Medical Center Research Ethics Committee has approved the study (MECID.No 2021723–10406), and informed consent was waived.

Radiological images were reviewed by two clinical specialists—one radiologist with seven years of experience in radiology and one pulmonologist with eleven years of experience in respiratory medicine. The radiological pattern and severity score were interpreted according to the research paper by Jin C et al. The radiological patterns were divided into three groups, namely bronchopneumonia, organising pneumonia, and diffuse alveolar damage pattern. The severity of radiological changes was scored according to the system as follows (CT score): 0 for 0% lobe involvement; 1 for 1–25% lobe involvement; 2 for 26–50% lobe involvement; 3 for 51–75% lobe involvement; 4 for 76–100% lobe involvement. A total severity score is calculated by summing the scores of the five lobes (range, 0–20) [10].

### Severe disease

The severe disease includes symptomatic pneumonia with hypoxia requiring oxygen supplements to maintain percutaneous arterial oxygen saturation ($SpO_2$) of more than 94% and critical illness which requires mechanical ventilation support (invasive or non-invasive) or vasopressor therapy [1].

### Persistent respiratory failure

This includes patients requiring oxygen therapy to maintain $SpO_2$ of more than 92% beyond 14 days or more of illness.

### Pulse corticosteroid

We considered a daily prednisolone-equivalent dose of more than 100 mg for at least 3 days as a pulse corticosteroid.

### Extended duration

Duration of corticosteroids of more than the recommended 10-day course is considered extended.

## Results

### Study population

In our cohort, the mortality rate among severe COVID-19 was 23.1% (173/749). Among the survivors, 46.2% (266/576) had persistent respiratory failure (PRF) after 14 days of illness. Out of these, 187 (70.3%) patients with high-resolution computed tomography (HRCT) done during admission came back for follow-up. Fig 1 shows the flow of patients included in the study.

### Basic characteristics and management

Table 1 demonstrates the sociodemographic, clinical characteristics and outcomes of PRF patients with or without prednisolone as a prolonged therapy. Table 2 summarises the

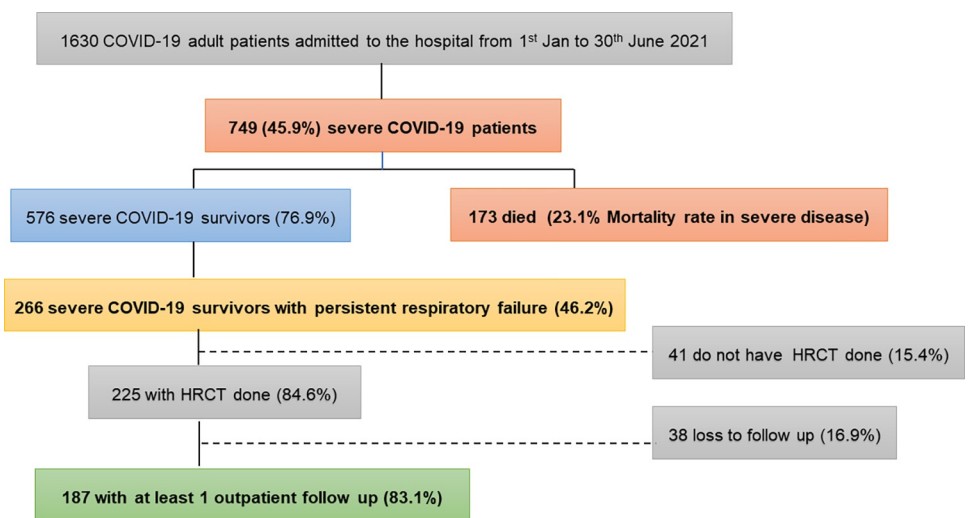

**Fig 1. Follow-up of severe COVID-19 patients with persistent respiratory failure.**

**Table 1. Sociodemographic and clinical characteristics and outcome of severe COVID-19 patients with persistent respiratory failure.**

| Characteristics | Overall, n:187 | Prednisolone group, n:128 | Non-prednisolone group, n:59 | P value |
|---|---|---|---|---|
| **Age, year** | 61.5 ± 12.6 | 62.1 ±12.4 | 60.1 ±13.0 | 0.296 |
| **Gender** | | | | 0.154 |
| Male | 116 (62) | 75(58.6) | 41(69.5) | |
| Female | 71 (38) | 53(41.4) | 18 (30.5) | |
| **Ethnicity** | | | | 0.788 |
| Malays | 101 (54) | 69 (53.9) | 32 (54.2) | |
| Chinese | 61 (32.6) | 44 (34.4) | 17 (28.8) | |
| Indian | 22 (11.8) | 13 (10.2) | 9 (15.3) | |
| Others | 3 (1.6) | 2 (1.6) | 1 (1.7) | |
| **Comorbidities** | | | | |
| Diabetes mellitus | 89 (47.6) | 59 (46.1) | 30 (50.8) | 0.545 |
| Hypertension | 105 (56.1) | 78 (60.9) | 27 (45.8) | 0.052 |
| Obesity (BMI> = 30) | 57 (30.5) | 40 (31.3) | 17(28.8) | 0.737 |
| CKD | 11 (5.9) | 8 (6.3) | 3 (5.1) | 1.000 |
| CVD | 27 (14.4) | 17 (13.3) | 10 (16.9) | 0.507 |
| Chronic lung diseases | 13 (7) | 8 (6.3) | 5 (8.5) | 0.758 |
| **Laboratory results upon admission** | | | | |
| CRP | 81.7 ± 70.1 | 84.0 ±70.4 | 76.7 ± 69.8 | 0.506 |
| Serum Ferritin | 1121 ± 963 | 1192 ± 992 | 968.2 ± 886.7 | 0.139 |
| ALC | 1.07 ± 0.53 | 1.08 ± 0.55 | 1.04 ± 0.49 | 0.709 |
| **Radiological pattern** | | | | **0.001** |
| Bronchopneumonia | 7 (3.7) | 1 (0.8) | 6 (10.2) | |
| Organising Pneumonia | 152 (81.3) | 103 (80.5) | 49 (83.1) | |
| Diffuse alveolar damage | 28 (15) | 24 (18.8) | 4 (6.8) | |
| **CT severity score** | 10 ± 3 | 11 ± 3 | 9 ± 3 | 0.001 |
| **Day of illness upon performing CT, median (IQR)** | 11 (9–14) | | | |
| **Maximum respiratory support** | | | | **0.004** |
| *High oxygen requirement* | 117 (62.6) | 89 (69.5) | 28 (47.5) | |
| Invasive mechanical ventilation | 28 (15) | 22 (17.2) | 6 (10.2) | |
| NIV | 1 (0.5) | 1 (0.8) | 0 (0) | |
| HFNC | 77 (41.2) | 60 (46.9) | 17 (28.8) | |
| NRM | 11 (5.9) | 6 (4.7) | 5 (8.5) | |
| *Low oxygen requirement* | 70 (37.4) | 39 (30.5) | 31 (52.5) | |
| 40–60% | 50 (26.7) | 30 (23.4) | 20 (33.9) | |
| <40% | 20 (10.7) | 9 (7.0) | 11 (18.6) | |
| **Complications** | | | | |
| Pulmonary embolism | 35 (18.7) | 28 (22) | 7 (11.9) | 0.098 |
| Hospital-acquired infection | 27 (14.4) | 20 (15.6) | 7 (11.9) | 0.497 |
| Diabetic emergency | 6 (3.2) | 4 (3.1) | 2 (3.4) | 1.000 |
| Gastro-intestinal bleeding | 5 (2.7) | 3 (2.3) | 2 (3.4) | 1.000 |
| **Length of hospital stays, day** | 20 ± 10 | 22 ±11 | 17 ±8 | **0.002** |
| **Short-term oxygen therapy upon discharge** | 27 (14.4) | 23 (18.0) | 4 (6.8) | **0.043** |
| **Resolution of >50% CXR lung opacity at follow-up* (6 weeks ±3)** | 133 (91.7) | 96 (92.3) | 37 (90.2) | 0.685 |

Abbreviation: Chronic kidney disease = CKD, cardiovascular diseases = CVD, BMI = Body mass index, c-reactive protein = CRP, absolute lymphocytes count = ALC, IQR = interquartile range, NIV = non-invasive ventilator, HFNC = high flow nasal cannula, NRM = non-rebreather mask.

Data are presented as n (%) or mean ± standard deviation unless otherwise stated.

*Only 146 patients were assessed.

**Table 2. Systemic corticosteroid and adjunctive therapy during acute phase of COVID-19 infection.**

| Treatment | Overall | Prednisolone group, n:128 | Non-prednisolone group, n:59 | P value |
|---|---|---|---|---|
| **Acute phase** | | | | |
| *Type of corticosteroid* | | | | |
| Methylprednisolone | 82 (43.9) | 70 (54.7) | 12 (20.3) | <**0.001** |
| Dexamethasone | 183 (97.9) | 125 (97.7) | 58 98.3 | 1.000 |
| Prednisolone | 22 (11.8) | 10 (7.8) | 12 (20.3) | **0.013** |
| *Prescribing pattern* | | | | |
| Pulse steroid | 93 (49.7) | 77 (60.2) | 16 (27.1) | <**0.001** |
| Extended duration of >10 days | 83 (44.4) | 47 (36.7) | 36 (61.0) | **0.002** |
| *Adjunctive treatment* | | | | |
| Tocilizumab | 87 (46.5) | 61 (47.7) | 26 (44.1) | 0.648 |
| Baricitinib | 15 (8.0) | 10 (7.8) | 5 (8.5) | 1.000 |
| Flavipiravir | 50 (26.7) | 38 (29.7) | 12 (20.3) | 0.179 |

Abbreviation: O2 = Oxygen. Data are presented as n (%) or mean ± standard deviation.

management of our study population. Sixty-eight percent (128/187) of them received prolonged corticosteroids with oral prednisolone. The mean starting dose of prednisolone was 0.68 mg/kg/day with a mean treatment duration of 47 days (SD±18).

## Radiological characteristics

The findings of HRCT were categorised into three groups (Fig 2).

## Follow up outcome

Thirty-six percent (67/187) of patients with PRF reported having persistent respiratory symptoms at 4 weeks follow-up (SD±2), with symptoms of shortness of breath on exertion being the commonest complaint.

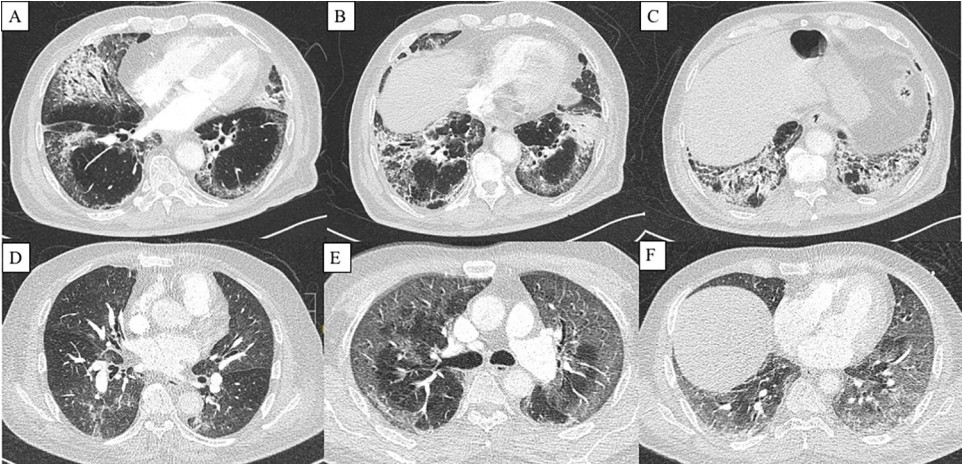

**Fig 2. Axial HRCT images.** (A, B, C) Organising pneumonia -bilateral bronchocentric consolidation over right middle lobe and left lower lobe extending to subpleural region with perilobular thickening. (D) Bronchopneumonia—multifocal peribronchial patchy consolidation and ground-glass changes. (E, F) Diffuse alveolar damage—diffuse peripheral ground-glass opacities and interlobular thickening.

Twelve patients had not achieved the expected radiological resolution. Among 12 patients with less than 50% resolution of CXR, four defaulted follow-ups, and five had complete resolution or residual changes within one year of follow-up. Two were diagnosed with pulmonary tuberculosis, and one had changes suggestive of interstitial lung disease. Only 10 patients had readmissions within 3 months. Among them, three were readmitted for COVID sequelae (one presented with pulmonary embolism and two presented with PRF). Four were readmitted for infections (two had bacteraemia, one had pulmonary tuberculosis, and one had catheter-related urinary tract infection). The other reasons for readmission were acute exacerbation of asthma and acute stroke (one patient in each diagnosis). One patient died in the casualty department for uncertain reasons.

As for functional assessment, 98/187 (52.4%) had a 6-minute walk test (6MWT), and 29/187 (15.5%) had spirometry done. The mean nadir $SpO_2$ was 92% (SD±5), and the mean distance was 338 meters (SD±104). The mean % predicted FEV1 was 81.6% (SD±21.4), and the mean % predicted FVC was 69.5% (SD±16.2).

## Comparison of two treatment groups

There were no statistically significant differences in age, gender, ethnicity, comorbidities, laboratory result upon admission and inpatient complications between those given and not given prednisolone as a prolonged therapy. Prednisolone was frequently prescribed in patients with OP and DAD radiological changes, higher CT severity scores, higher oxygen requirements, and in those given pulse corticosteroids, extended corticosteroids treatment during the acute phase, and individuals who require short-term oxygen therapy upon discharge. The mean length of hospitalisation was longer in patients who received prolonged prednisolone therapy. (P value = 0.002) There was no statistically significant radiological improvement between these groups upon outpatient follow-up. (P value = 0.685).

## Discussion

Our study observed the disease course and clinical outcome of COVID-19 survivors with PRF. A vast majority of patients had OP pattern as the radiological findings. They were much more likely to be given prolonged corticosteroid therapy in light of the evidence of corticosteroid responsiveness in both cryptogenic organising pneumonia and secondary organising pneumonia [11, 12]. The prednisolone group was more likely to be critically ill with higher CT severity scores and oxygen requirements, be given pulse corticosteroids during the acute phase, have a longer residual disease course, and require short-term oxygen therapy upon discharge. Nonetheless, we had a high follow-up rate, and many patients had significant clinical improvement at 6 ±3 weeks of follow-up.

The predominant OP radiological pattern was similar to the study by Myall et al. (59%) [7]. However, our OP rate was higher in light of a different study cohort. OP is not uncommon, as it was reported in up to 34–44% of 100 autopsy findings of multiple centres. However, the most prevalent finding is still diffuse alveolar damage (DAD), which ranges from 75–87% [13, 14]. The timing of performing the post-mortem may affect the pathological findings, as studies have shown that DAD pathology is the predominant feature within the first 2 weeks of disease, whereas OP pathology was noted at a later course of the disease, which is around 20 days of disease [14, 15]. Secondary OP is also seen in other infections, such as pneumococcal pneumonia, atypical pneumonia, viral pneumonia (including adenovirus, cytomegalovirus, influenza, parainfluenza, HIV, etc), parasitic and fungal pneumonia [12]. As lung biopsy is an invasive procedure, we felt that performing such a test is not feasible in patients who are recuperating from a critical condition, as this may pose a potential risk of pneumothorax that can further

worsen their respiratory failure. Therefore, none of our patients had a biopsy-proven OP, apart from the radiological pattern. Radiological changes consistent with organising pneumonia have already been reported during the early pandemic in China [16]. Alongside this, a narrative review of case reports and series by Chong et al. has pointed out the existence of pathological and radiological proven secondary OP in COVID-19. The worsening symptoms after the cessation of corticosteroids are seen, which rendered a prolonged corticosteroid to achieve remission [17].

OP is a major repairing process of the pulmonary tissue after an insult, manifesting as an organisation of inflammatory exudate in the air alveoli by the fibrous tissue. This intra-alveolar fibrosis has been proven to be responsive to corticosteroids by its intriguing characteristic of inhibiting and reversing fibrosis formation [12]. In contrast, the effectiveness of corticosteroids is not seen in DAD at any stage [18]. Some of our DAD patients were given prolonged prednisolone therapy, and this might not be beneficial.

On the other hand, E. Lappi-Blanco et al. demonstrated that vascular endothelial growth factor and basic fibroblast growth factor are found abundantly in intraluminal fibromyxoid tissue of OP and hence promoting angiogenesis and reversal of the fibrosis [19]. Therefore, spontaneous resolution of OP without corticosteroid is also attainable. Although the role of systemic corticosteroids in acute COVID-19 has been well-established, a meta-analysis by Li et al. has demonstrated the association between corticosteroids and prolonged viral shedding (PVS), especially in the medium to high-dose corticosteroid group [20–22]. Whether PVS is the cause of prolonged hospitalisation was not explored in our study.

The exact incidence of reactivation of tuberculosis in post-COVID-19 patients is not known apart from being described in the case series. It is relevant as Malaysia is a moderate-to-low endemic country for tuberculosis [23]. Previous use of corticosteroids is a known risk for this phenomenon, and this poses a notably higher risk if the daily corticosteroid dose is more than 15 mg of prednisolone or equivalent for a duration of 2 to 4 weeks [24]. Mucormycosis is a devastating secondary infection in COVID-19 reported in India, which is associated with corticosteroid use and diabetes mellitus [25]. Interestingly, we have not observed any similar case in our trust.

With limited healthcare resources to handle the sudden high demand of patients with persistent respiratory failure, short-term oxygen therapy (STOT) or home oxygen therapy was inevitably used to decongest the healthcare system, despite the lack of evidence on such practice during the COVID-19 pandemic. Careful assessment before prescription of STOT is crucial to ascertain its safety. Life-threatening conditions or other contributing factors of hypoxaemia, such as congestive heart failure, pulmonary embolism, superimposed pneumonia, or alveolar hypoventilation syndrome, must be excluded before treatment decision [26, 27]. About 14.4% of our patients who were experiencing PRF solely secondary to COVID-19 infection were stable enough to be discharged with STOT, and this corresponds to the 15.5% in our cohort of severe COVID-19 who were mechanically ventilated. Similarly, as described by Musheyev et al., the incidence of persistent respiratory failure is high in mechanical ventilated COVID-19 survivors, with 50% of them needing oxygen supplementation upon discharge [28].

The retrospective design is the main limitation of our study. Some patients lost to follow-up, which contributed to the missing data, and not all the clinical parameters were assessed during the follow-up. Hence, we were unable to make a meaningful comparison between prednisolone and non-prednisolone groups. There was also variable timing of HRCT performed from the day of illness and clinic evaluation from the day of discharge. However, our follow-up rate was higher than other studies, as we have a multidisciplinary COVID clinic established specifically to cater to this particular group of patients. Another limitation is that our study

was done prior to the circulation of the Omicron variant of SARS-CoV-2, which is known to have a shorter duration and lower severity of the disease. Some study reveals less typical radiological changes compared to the Delta variant [29]. The current high seroprevalence of humoral immune responses toward COVID-19 from previous natural infections and vaccination may have also changed the clinical course of COVID-19.

Our study is the first to describe the disease course, radiological pattern and clinical outcome of severe COVID-19 with PRF in two different groups: with or without prolonged prednisolone therapy. One size does not fit all, given the heterogenicity of COVID-19 infection. It is crucial to delineate criteria for patients who may benefit from prolonged prednisolone therapy based on clinical characteristics and radiological patterns. Patients may benefit from the prednisolone treatment and probably prevent deleterious sequelae, yet to be confirmed in the literature after weighing balance with its long-term side effects. This study may provide a framework for future studies exploring the role of prolonged corticosteroid in certain patients.

## Conclusion

Almost half of the patients with severe COVID-19 had PRF, with a predominant radiological OP pattern. More than two-thirds of the PRF patients required prolonged oral corticosteroid treatment. Familiarising clinicians with the disease course, radiological patterns, and potential outcomes of this group of patients may better equip them to manage their patients.

## Supporting information

**S1 File.**
(XLSX)

## Acknowledgments

We would like to extend our heartfelt gratitude to our patients, who have been our greatest source of inspiration, and our fellow colleagues who are joining us in this fight.

## Author Contributions

**Conceptualization:** Chee Kuan Wong, Leng Cheng Sia.

**Data curation:** Leng Cheng Sia, Noreen Zhi Min Ooi.

**Formal analysis:** Chee Kuan Wong, Leng Cheng Sia.

**Funding acquisition:** Chee Kuan Wong.

**Investigation:** Leng Cheng Sia.

**Methodology:** Chee Kuan Wong, Leng Cheng Sia.

**Project administration:** Chee Kuan Wong, Leng Cheng Sia.

**Resources:** Chee Kuan Wong, Leng Cheng Sia, Noreen Zhi Min Ooi, Wai Yee Chan.

**Software:** Leng Cheng Sia, Wai Yee Chan.

**Supervision:** Chee Kuan Wong.

**Validation:** Chee Kuan Wong, Leng Cheng Sia, Noreen Zhi Min Ooi, Wai Yee Chan, Yongkek Pang.

**Visualization:** Leng Cheng Sia, Noreen Zhi Min Ooi, Wai Yee Chan.

**Writing – original draft:** Chee Kuan Wong, Leng Cheng Sia.

**Writing – review & editing:** Chee Kuan Wong, Leng Cheng Sia, Noreen Zhi Min Ooi, Wai Yee Chan, Yong-kek Pang.

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
