## [Decision Letter · Decision Letter 0]

3 Aug 2022

PONE-D-22-15286Severe COVID-19 with persistent respiratory failure- A retrospective cohort study in a tertiary centre in Malaysia.PLOS ONE

Dear Dr. Sia,

Thank you for submitting your manuscript to PLOS ONE. After careful consideration, we feel that it has merit but does not fully meet PLOS ONE’s publication criteria as it currently stands. Therefore, we invite you to submit a revised version of the manuscript that addresses the points raised during the review process.

We suggest to focus the discussion section on the novelty of the present investigation. Results should not be repeated in the Discussion, but evidence on this topic should be cited. The reason(s) why corticosteroids may or may have not been efficaceous shoudl be better explained. 

We look forward to receiving your revised manuscript.

Kind regards,

Chiara Lazzeri

Academic Editor

PLOS ONE

Journal Requirements:

Reviewers' comments:

Reviewer's Responses to Questions

**Comments to the Author**

1. Is the manuscript technically sound, and do the data support the conclusions?

Reviewer #1: Partly

Reviewer #2: Yes

2. Has the statistical analysis been performed appropriately and rigorously? 

Reviewer #1: No

Reviewer #2: Yes

3. Have the authors made all data underlying the findings in their manuscript fully available?

Reviewer #1: No

Reviewer #2: Yes

4. Is the manuscript presented in an intelligible fashion and written in standard English?

Reviewer #1: Yes

Reviewer #2: Yes

5. Review Comments to the Author

Reviewer #1: The present paper reported the retrospective study of severe COVID-19 patients who were admitted to a single tertiary centre from 1 January 2021 till 30 June 2021. Further, the study mainly focused on the persistent respiratory failure (PRF).

Among the patients who came for follow up all were in higher age group (mean 61.5+12.6). Of these patients 56.1% had hypertension followed by diabetes mellitus (47.6%), obesity (30.5%), CVD (14.4%), CKD (5.9%). The authors didn’t try to find any relation of these co-morbidities with the PRF and prolonged treatment of oral corticosteroid.

In addition, the authors have accepted in the study that they were unable to conclude that the use of oral corticosteroid maintenance has its beneficial effect in preventing the inflammatory sequelae, or if supportive care alone with oxygen support is sufficient to help in the process of patient’s recovery.

In conclusion this paper is not giving any conclusive findings for the usage corticosteroid for long term to the patients with severe COVID-19.

Reviewer #2: Very nice descriptive case series describing duration of organizing pneumonia in COVID19 survivors in a Malaysian hospital showing prolonged time to recovery, prolonged use of corticosteroids, and some TB reactivation and secondary infection.

6. PLOS authors have the option to publish the peer review history of their article (what does this mean?). If published, this will include your full peer review and any attached files.

Reviewer #1: No

Reviewer #2: No

---

## [Author Response · Author response to Decision Letter 0]

10 Oct 2022

Editor:

We thank you for the suggestion. We have made the changes to focus more on the novelty of findings by elaborating more on OP and the reasons of corticosteroid may or may not help in COVID-19 patients with PRF with two additional references [18,19] We also discussed more on the likely harmful effect of corticosteroid with opportunistic infection which has been reported in the literature. Apart from this, we have removed the repetition of result such as number of PRF, patients on prolonged corticosteroid with prednisolone and follow up outcome from the first paragraph of discussion. 

We have changed the reference brackets to square brackets to meet the requirements. 

Knowing the importance of minimal data set to be accessible to all the readers, we have uploaded the full dataset for the study population in Supporting Information file with identities being anonymised. 

We have also added University Malaya Medical Centre ethics committee approval statement in method section. 

Reviewer 1:

We truly appreciate the comment which has stimulated more analysis with some new findings. We have included the bivariate analysis between patient PRF with or without a prolonged treatment of oral corticosteroid and noted that there were no statistically significant differences in age, gender, ethnicity, comorbidities, laboratory result upon admission and inpatient complications between the two different groups. It was noted that prednisolone was frequently prescribed in patients with OP and DAD radiological changes, higher CT severity score, high requirement of oxygen, given pulse steroid or methylprednisolone, extended duration corticosteroid during acute treatment, longer hospital stays and given short term oxygen therapy upon discharge, There was no statistically significant of radiological improvement between these groups upon outpatient follow up.

We agree that retrospective design is difficult to conclude the benefit of corticosteroid in some group of patients. Our study was to show OP may be the cause of PRF in some patients and hopefully it can provide a framework for future study to explore the benefit of prolonged corticosteroid in certain group of patients as being shown in some case series and observational study. Prospective study is required to investigate any benefit of prolonged steroid in patients with PRF and OP changes. 

Reviewer 2:

We thank the acknowledgment from the reviewer. 

Overall, we have incorporated all the suggestions into the revision. Thanks for the help.

---

## [Editor Report · Decision Letter 1]

17 Oct 2022

Severe COVID-19 with persistent respiratory failure- A retrospective cohort study in a tertiary centre in Malaysia.

PONE-D-22-15286R1

Dear Dr. Sia,

We’re pleased to inform you that your manuscript has been judged scientifically suitable for publication and will be formally accepted for publication once it meets all outstanding technical requirements.

Kind regards,

Chiara Lazzeri

Academic Editor

PLOS ONE
---

## [Editor Report · Acceptance letter]

26 Oct 2022

PONE-D-22-15286R1 

Severe COVID-19 with persistent respiratory failure - A retrospective cohort study in a tertiary centre in Malaysia. 

Dear Dr. Sia:

I'm pleased to inform you that your manuscript has been deemed suitable for publication in PLOS ONE. Congratulations! Your manuscript is now with our production department. 

Kind regards, 

on behalf of

Dr. Chiara Lazzeri 

Academic Editor

PLOS ONE